# Dynamics-Augmented Decision Transformer for Offline Dynamics Generalization

**Changyeon Kim**[*,1]    **Junsu Kim**[*,1]    **Younggyo Seo**[1]
**Kimin Lee**[2]    **Honglak Lee**[3,4]    **Jinwoo Shin**[1]

[1]KAIST    [2]Google Research    [3]University of Michigan    [4]LG AI Research

## Abstract

Recent progress in offline reinforcement learning (RL) has shown that it is often possible to train strong agents without potentially unsafe or impractical online interaction. However, in real-world settings, agents may encounter unseen environments with different dynamics, and generalization ability is required. This work presents Dynamics-Augmented Decision Transformer (DADT), a simple yet efficient method to train generalizable agents from offline datasets; on top of return-conditioned policy using the transformer architecture, we improve generalization capabilities by using representation learning based on next state prediction. Our experimental results demonstrate that DADT outperforms prior state-of-the-art methods for offline dynamics generalization. Intriguingly, DADT without fine-tuning even outperforms fine-tuned baselines.

## 1 Introduction

Offline reinforcement learning (RL) provides a framework to train robotic agents from a logged dataset without online interaction (Lange et al., 2012; Levine et al., 2020). The benefit of offline RL enables us to utilize RL for complex robotics domains where environment interactions can be expensive or dangerous (Dulac-Arnold et al., 2019). However, for practical applications, besides the ability to learn without interaction, the ability to generalize to unseen environments with (slightly) different dynamics is required. Unfortunately, it has been evidenced that offline RL often struggles to generalize to unseen dynamics (Li et al., 2020; Lin et al., 2022).

Several approaches have been proposed recently to address dynamics generalization in offline RL, including representation learning for dynamics inference (Li et al., 2020) and meta-learning (Lin et al., 2022). Specifically, Li et al. (2020) trains a context encoder for efficient dynamics inference on top of offline RL, and Lin et al. (2022) learns meta-dynamics model and meta-policy that adapt to unseen dynamics in a few gradient steps. However, whether a simple gradient update or context latent variable from near past experiences could capture subtle changes in differing dynamics is questionable, which is essential for solving dynamics generalization problems.

Turning to vision and language domains, transformer-based models (Vaswani et al., 2017) have exhibited impressive results in solving various generalization problems. As a representative example, Brown et al. (2020), Rae et al. (2021), and Chowdhery et al. (2022) show that large language model using transformer architecture can solve various types of unseen tasks such as translation, question answering, even without any demonstration. Ramesh et al. (2021, 2022) demonstrate that transformer-based models jointly trained with the text and image tokens can generate photorealistic images from unseen short-length captions. Recently, Reed et al. (2022) show that transformers can be a generalist agent that can solve a range of control tasks using inputs from different modalities with a single set of weights.

---

[*]Equal Contribution. Correspondence to `changyeon.kim@kaist.ac.kr`.

Offline Reinforcement Learning Workshop at Neural Information Processing Systems, 2022

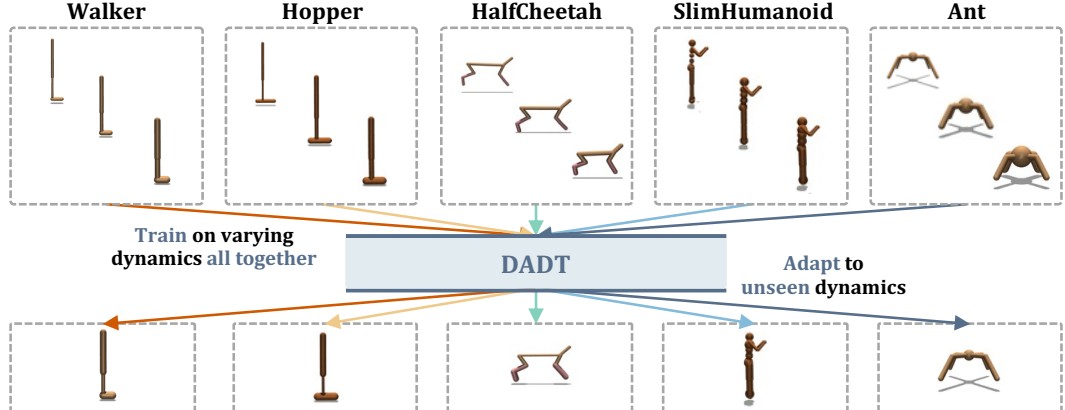

Figure 1: Illustration of **D**ynamics-**A**ugmented **D**ecision **T**ransformer (DADT). During training, DADT digests offline trajectories from varying dynamics without additional annotation (i.e., dynamics id). We evaluate DADT on unseen test dynamics and DADT shows strong generalization performance.

**Contribution.** Inspired by the generalization ability of transformer-based models in vision (Ramesh et al., 2022; Dosovitskiy et al., 2020) and language domains (Brown et al., 2020; Devlin et al., 2018), we present DADT: **D**ynamics-**A**ugmented **D**ecision **T**ransformer, a new transformer-based method for offline dynamics generalization. Analogously to pre-training scheme in transformer-based models in vision and language domains, we train DADT via a supervised learning paradigm with totally aggregated sequences (i.e., trajectories) without distinction. Specifically, DADT aims to optimize the objective to find the action that achieves desired return (Chen et al., 2021) while minimizing error in next state prediction. We find that DADT can better generalize to unseen dynamics by explicitly encouraging the model to learn dynamics information. For evaluation, we consider two scenarios depending on different accessibility to trajectories from unseen dynamics: (1) zero-shot, where demonstrations are not given, and (2) adaptation, where model is fine-tuned with small amount of samples. We provide an overview of setup and architecture in Figure 1 and Figure 2, respectively.

We show that DADT outperforms prior state-of-the-art baselines for offline dynamics generalization in both performance and time efficiency on various continuous control tasks based on MuJoCo simulator (Todorov et al., 2012), which is widely used in the literature (Li et al., 2020; Lin et al., 2022). Intriguingly, DADT without fine-tuning even outperforms fine-tuned baselines. Moreover, reduced next state prediction error in unseen dynamics further supports our model's capability for dynamics generalization.

## 2 Related Work

### 2.1 Dynamics generalization and adaptation

Recent works handling dynamics generalization focus on encoding inductive bias using prior knowledge (Zambaldi et al., 2018) or learning contextual latent encoder that captures local dynamics (Lee et al., 2020; Seo et al., 2020). Apart from this, several meta-learning approaches to dynamics generalization have been proposed for quickly adapting to new tasks (Gupta et al., 2018; Rakelly et al., 2019; Zintgraf et al., 2019). Rakelly et al. (2019) introduced probabilistic context variable accumulating past experiences for efficient exploration and fast task inference. Recently, Melo (2022) uses transformer architecture for capturing contextual information instead of recurrent neural network Duan et al. (2016) in online dynamics generalization. It differs from our work in that (i) we use transformer architecture not only for capturing contextual information but also for optimizing policy conditioned on desired return, and (ii) our model utilizes offline datasets.

Along with increasing attention toward offline RL (Levine et al., 2020) of which importance is highlighted especially when environmental interaction is expensive or risky, dynamics generalization and adaptation in the offline setting have been studied recently (Li et al., 2020; Lin et al., 2022; Ball et al., 2021; Cang et al., 2021). They aim to learn a policy or dynamics model that can quickly

adapt to unseen dynamics along with mitigating value overestimation issues in offline RL (Fujimoto et al., 2019). Specifically, FOCAL (Li et al., 2020) trains a deterministic context encoder for efficient dynamics inference using contrastive loss and use it for behavior regularized policy learning (Wu et al., 2019). MerPO (Lin et al., 2022) takes a gradient-based meta-learning approach for learning meta-dynamics model and meta-policy that aim to adapt to unseen dynamics quickly; the meta-dynamics model is used to generate synthetic rollouts to boost policy learning. Ball et al. (2021) augment a learned dynamics model to improve the zero-shot generalization in model-based offline RL.

## 2.2 RL via sequence modelling

Motivated by remarkable success in large-scale language modeling (Devlin et al., 2018; Brown et al., 2020), it has been proposed to extend such success into RL domain by formulating RL as a sequence modeling problem (Chen et al., 2021; Janner et al., 2021). These works build on the reinforcement learning as supervised learning paradigm (Schmidhuber, 2019; Srivastava et al., 2019; Emmons et al., 2021) that focuses on predictive modeling of sequence of states, actions, and reward-based returns. Specifically, Chen et al. (2021) proposed Decision Transformer (DT), which is trained to generate the optimal actions given desired return and history of states and actions. Instead of conditioning desired return, Trajectory Transformer (TT) (Janner et al., 2021) is trained to predict not only actions, but also entire elements of trajectories such as states and rewards and finds optimal actions by using planning (e.g., beam search) in deployment time. Recently, Lee et al. (2022) have demonstrated that DT can play multiple Atari games with a single set of weights and can be efficiently fine-tuned for solving unseen games.

Though DT and TT have shown promising results in solving various control tasks, their ability to generalize to unseen dynamics has not investigated yet. In this paper, we demonstrate that DT trained on aggregated trajectories collected from different dynamics can generalize to unseen dynamics, and introducing an auxiliary next state prediction loss can further improve the performance.

## 3 Preliminaries

### 3.1 Problem statement

We consider the standard RL framework where an agent interacts with its environment in discrete time. Formally, we formulate our problem as a Markov decision process (MDP; Sutton & Barto (2018)), which is defined as a tuple $(\mathcal{S}, \mathcal{A}, p, r, \gamma, \rho_0)$. Here, $\mathcal{S}$ is the state space, $\mathcal{A}$ is the action space, $p(s'|s, a)$ is the transition dynamics, $r(s, a)$ is the reward function, $\rho_0$ is the initial state distribution, and $\gamma \in [0, 1]$ is the discount factor. A trajectory is made up of a sequence of states, actions, and rewards: $\tau = (s_0, a_0, r_0, s_1, a_1, r_1, \ldots, s_T, a_T, r_T)$, where $T$ is the (pre-defined) horizon. The return of a trajectory, $R(\tau) = \sum_{t=0}^{T} \gamma^t r_t$, is the sum of discounted future rewards. The goal of RL is to find a policy that maximizes the expected return $\mathbb{E}[R(\tau)]$. In our work, we focus on offline RL setting where we only have access to some fixed dataset consisting of trajectory rollouts of arbitrary policies instead of obtaining data via environment interaction.

In order to address the problem of generalization, we further consider the distribution of MDPs, where the transition dynamics $p_c(s'|s, a)$ varies according to a context $c$. For example, transition dynamics can vary across different terrains for a walking robot due to different amounts of friction. We aim to learn a single policy $\pi$ that maximizes the expected return robustly to such dynamics changes. To be specific, given trajectories from training dynamics with contexts sampled from $p_{\texttt{train}}(c)$, we aim at learning a policy $\pi$ that can maximize expected return for test dynamics with unseen (but related) contexts sampled from $p_{\texttt{test}}(c)$.

### 3.2 Transformer

Transformer (Vaswani et al., 2017) is an architecture to efficiently model sequential data by using stacked self-attention module with residual connection. Each self-attention module takes $n$ embeddings $\{x_i\}_{i=1}^n$ and outputs another $n$ embeddings $\{z_i\}_{i=1}^n$ with the same dimensions. Each token $x_i$ is projected to a key $k_i$, query $q_i$, and value $v_i$ using linear transformation. Then, the output $z_i$ is computed by weighting values $v_j$ by the normalized inner product between the query $q_i$ and other

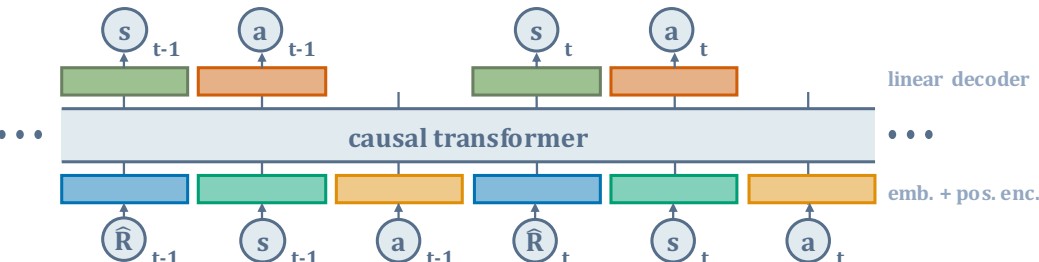

Figure 2: An overview of DADT architecture. Along with action prediction, we also predict next state for each timestep to boost dynamics generalization. DADT consumes trajectories from varying dynamics without any distinction, which is analogous manner in training language model; language tasks are aggregated without any task annotation in training. States $s_t$, actions $a_t$, and return-to-go $\hat{R}_t$ are fed into modality-specific linear embeddings and a positional timestep embedding is added.

keys $k_j$:

$$z_i = \sum_{j=1}^{n} \texttt{softmax}(\{\langle q_i, k_{j'} \rangle_{j'=1}^{n}\})_j \cdot v_j. \tag{1}$$

The self-attention modules work by performing the inner product between the current embedding $x_i$ with the query matrix $Q$ to give $q_i$ and with the key matrix $K$ to give the key $k_i$. The model takes the inner product between the key and query for $j \in [1, n]$, yielding $n$ logits, over which the model takes the softmax and yields a probability distribution. The final result is then a convex combination of the value vectors $v_i$, with the weights dictated by the probability distribution. More concisely, the $i$-th component of the output $z_i$ is given by

$$z_i = \sum_{j=1}^{n} \texttt{softmax}(\{\langle q_i, k_{j'} \rangle_{j'=1}^{n}\})_j \cdot v_j. \tag{2}$$

## 4 DADT: Dynamics-Augmented Decision Transformer

Following Chen et al. (2021), we consider the problem of offline reinforcement learning as a sequence modeling problem; we model the probability of the next sequence token $\tau_i$ conditioned on all tokens prior to it: $P_\theta(\tau_i | \tau_{<i})$. The sequences we consider have the form:

$$\tau = (\hat{R}_0, s_0, a_0, \hat{R}_1, s_1, a_1, \ldots, \hat{R}_T, s_T, a_T), \tag{3}$$

where $T$ denotes the max timestep, and $\hat{R}_t$ is the *return-to-go*, which is the sum of return for the rest of the sequence.

**Architecture.** we largely follow the architecture from Chen et al. (2021) and then append next state prediction head, which will be explained below. Specifically, our transformer takes the last $K$ timesteps as an input; namely, $3K$ tokens consisting of return-to-go, state, and action are fed. Raw inputs are projected into the embedding dimension by a linear layer for each modality followed by layer normalization (Ba et al., 2016). Additionally, we learn positional embedding per timestep and add it to each token; one timestep corresponds to three tokens. The tokens are then processed by a *causal* transformer, followed by action prediction head; future action tokens are predicted via autoregressive modeling along with masking future tokens in an input sequence. Analogously to the action prediction head, we stack a linear decoder for next state prediction on top of the transformer. We provide an illustration of our architecture in Figure 2.

**Training with next state prediction.** Along with the action prediction proposed by Chen et al. (2021), we train our DADT using next state prediction as an auxiliary training task to boost dynamics generalization. First, given a dataset of offline trajectories from varying dynamics, we sample mini-batches of sequence length $K$ from the dataset. Note that we do not include any dynamics-id in input sequences; this can be considered as a more practical setting because annotating such ids would require additional labeling costs in practice. For the total loss, we add action prediction loss and

Table 1: Normalized average return on test dynamics with episode length 200 across 5 runs. We mark the scores within one standard deviation from the highest average score to be bold. For FOCAL (Li et al., 2020), we only report the zero-shot performance because it is designed to be fully-offline without adaptation to unseen dynamics (see Section 5.1 for details).

| Environment | Evaluation | FOCAL (Li et al., 2020) | MerPO (Lin et al., 2022) | DADT (Ours) |
|---|---|---|---|---|
| Walker | Zero-shot | $52.44_{\pm 17.31}$ | $40.67_{\pm 6.66}$ | $\mathbf{61.12}_{\pm 4.99}$ |
| | Adaptation | - | $48.11_{\pm 6.95}$ | $\mathbf{64.65}_{\pm 4.42}$ |
| Hopper | Zero-shot | $41.56_{\pm 12.44}$ | $53.21_{\pm 1.89}$ | $\mathbf{74.69}_{\pm 4.22}$ |
| | Adaptation | - | $57.38_{\pm 6.24}$ | $\mathbf{83.11}_{\pm 3.28}$ |
| HalfCheetah | Zero-shot | $10.60_{\pm 4.10}$ | $8.06_{\pm 1.09}$ | $\mathbf{16.29}_{\pm 1.95}$ |
| | Adaptation | - | $14.36_{\pm 3.27}$ | $\mathbf{23.48}_{\pm 3.39}$ |
| SlimHumanoid | Zero-shot | $20.45_{\pm 1.60}$ | $33.73_{\pm 1.41}$ | $\mathbf{42.03}_{\pm 1.68}$ |
| | Adaptation | - | $32.51_{\pm 1.52}$ | $\mathbf{39.58}_{\pm 1.79}$ |
| Ant | Zero-shot | $18.73_{\pm 1.28}$ | $31.34_{\pm 0.80}$ | $\mathbf{35.87}_{\pm 1.62}$ |
| | Adaptation | - | $37.08_{\pm 0.79}$ | $\mathbf{54.85}_{\pm 1.81}$ |

auxiliary next state prediction loss, multiplied by balancing coefficient $\lambda$. This next state prediction loss enriches the training signal, which can help capture subtle changes across varying dynamics.

While our next state prediction loss has a connection with that of Trajectory Transformer (Janner et al., 2021) in that both of them predict next state given past transitions, we remark that DADT does not use discretization, which might be a negative factor in dynamics generalization. Specifically, approximating a set of values from different dynamics using the exact discrete quantities could hurt capturing subtle changes in distinguishing the difference between dynamics.

**Evaluation.** When we encounter unseen dynamics, we evaluate both zero-shot generalization and few-shot adaptation ability of DADT. For zero-shot, we do not update our DADT for the new dynamics assuming that trajectories from the new environments are unavailable. For few-shot adaptation, we fine-tune DADT in a supervised manner given a fixed number of samples from the new environments, which is the same as in the training phase. For deployment, we specify the desired performance by conditioning return-to-go $\hat{R}_0$ along with the starting state $s_0$. Then, we generate action for the current state (in a deterministic manner). After feeding the generated action, we decrement the target return by the achieved reward and repeat until episode termination.

## 5 Experiments

### 5.1 Setups

**Environments.** We demonstrate the effectiveness of our model in various simulated robotic control environments (i.e., Walker, Hopper, HalfCheeatah, SlimHumanoid, Ant) based on the OpenAI Gym (Brockman et al., 2016). In each environment, we consider 20 training dynamics and 5 test dynamics with a different set of dynamics parameters (mass, inertia, damping, friction) following setups of Finn et al. (2017) and Rakelly et al. (2019). Note that test dynamics have different dynamics parameters from training dynamics. We provide detailed descriptions of our environments in Appendix A.

**Offline data collection.** We follow the same procedure for offline data collection in Li et al. (2020). Specifically, we train policies using soft actor-critic (Haarnoja et al., 2018) for every single dynamics and collect trajectories every 50K timesteps. Collected trajectories have diverse qualities; trajectories from early timestep are likely to have lower average return, while later ones have higher return. For collecting each trajectory, we set the max horizon $T$ as 200 following Li et al. (2020) and Lin et al. (2022). We provide detailed statistics of collected offline datasets in Appendix B.

**Implementation details.** We largely follow the implementation details of the original Decision Transformer (Chen et al., 2021). Specifically, we train DADT for 1M steps using AdamW optimizer (Loshchilov & Hutter, 2017) with a learning rate of $3 \times 10^{-5}$ containing linear warmup steps of 10K, weight decay of $10^{-4}$, gradient clip of 1.0, and batch size of 64. For fine-tuning DADT on unseen

Table 2: Normalized average return on test dynamics with episode length 200 in zero-shot scenario across 5 runs. We mark the scores within one standard deviation from the highest average score to be bold. Transformer-BC trains transformer using behavior cloning without return-to-go conditioning.

| | Walker | Hopper | HalfCheetah | SlimHumanoid | Ant |
|---|---|---|---|---|---|
| Transformer-BC | 46.50 $_{\pm 7.66}$ | 62.63 $_{\pm 7.25}$ | 16.31 $_{\pm 2.87}$ | 38.87 $_{\pm 3.94}$ | 34.37 $_{\pm 0.48}$ |
| DT (Chen et al., 2021) | **59.81** $_{\pm 6.30}$ | **71.49** $_{\pm 4.66}$ | **20.86** $_{\pm 2.30}$ | 39.93 $_{\pm 3.62}$ | 32.04 $_{\pm 2.28}$ |
| DADT (Ours) | **61.12** $_{\pm 4.99}$ | **74.69** $_{\pm 4.22}$ | 16.29 $_{\pm 1.95}$ | **42.03** $_{\pm 1.68}$ | **35.87** $_{\pm 1.62}$ |

dynamics, we train DADT with a learning rate of $10^{-5}$, weight decay of $10^{-2}$, and batch size of 32. We provide further implementation details in Appendix C.

**Evaluation protocol.** We train DADT and baselines with 5 independent runs and evaluate them across 5 (unseen) test dynamics. To assess performance in the test dynamics, we measure returns of 10 rollouts whose max timestep is 200. Then, to facilitate comparison across different environments, we normalize returns for each dynamics to the range between 0 to 100, by computing $\texttt{Normalized\_return} = 100 \times \frac{\texttt{return} - \texttt{random\_return}}{\texttt{expert\_return} - \texttt{random\_return}}$ following setups in Chen et al. (2021) and Fu et al. (2020). For $\texttt{random\_return}$ and $\texttt{expert\_return}$, we use minimum/maximum return of collected trajectories, respectively, which are collected periodically while training a soft actor-critic (Haarnoja et al., 2018) for each test dynamics. Finally, we average out 5 (runs) $\times 5$ (test dynamics) $\times 10$ (rollouts) number of returns to measure performance. For DADT and baselines, we measure the test performance periodically during training and report the best return.

**Baselines.** To evaluate the performance of our method, we consider the following state-of-the-art methods for offline dynamics generalization:

- **FOCAL** (Li et al., 2020): an offline model-free actor-critic meta-RL method with (i) a deterministic context encoder for efficient task inference and (ii) behavior regularization (Wu et al., 2019) to mitigate bootstrapping errors from offline dataset. We only evaluate FOCAL in a zero-shot setup because FOCAL is designed to be fully offline without adaptation to unseen dynamics. In particular, context encoder in FOCAL is trained with distance metric loss (Sohn, 2016) that pushes away samples from other dynamics in the embedding space. This makes it non-trivial to fine-tune the model using samples from a single unseen dynamics. A comparison with this method shows that DADT could capture the context of different dynamics without contrastive learning.

- **MerPO** (Lin et al., 2022): an offline model-based meta-RL method in which proximal meta-learning (Zhou et al., 2019) is used for training both task-specific dynamics model and task-specific actor-critic networks. We report evaluation results (i) using meta-policy and meta-dynamics model without adaptation (zero-shot) and (ii) using task-specific policy and dynamics model after adaptation to unseen dynamics (adaptation). A comparison with this method shows that simple supervised learning scheme with transformer-based model could perform comparably or better than complex meta-learning approaches (Finn et al., 2017; Stadie et al., 2018; Rothfuss et al., 2018; Mitchell et al., 2021).

## 5.2 How does DADT perform compared to prior state-of-the-art methods?

We compare DADT to the prior state-of-the-art methods in two scenarios: zero-shot and adaptation. As shown in Table 1, DADT surpasses the prior arts across all the environments and evaluation scenarios. We also find that DADT without any adaptation outperforms adapted MerPO in 4 out of 5 environments, which further highlights the dynamics generalization capability of DADT. We also emphasize that DADT does not use any dynamics annotation in training, while other baselines are trained with trajectories annotated with dynamics-id.

## 5.3 How does DADT perform compared to DT and BC?

To investigate the effect of the proposed auxiliary next state prediction for improving dynamics generalization, we compare DADT with DT in Table 2 in a zero-shot setup, where the same network architecture and hyperparameters are used for both DADT and DT. We find that DADT outperforms

Table 3: Aggregated prediction error of DT+ and DADT on test dynamics across 5 runs. We mark the scores with one standard deviation from the lowest prediction error to be bold. DT+ is a variant of DT, which is trained using only action prediction loss, and then the additional linear layer for state prediction is stacked and trained on top of the (frozen) trained DT.

|  | Walker | Hopper | HalfCheetah | SlimHumanoid | Ant |
|---|---|---|---|---|---|
| DT+ | 0.2314 $\pm 0.0050$ | 0.3346 $\pm 0.0115$ | 0.8001 $\pm 0.0157$ | 0.4766 $\pm 0.0113$ | 0.6160 $\pm 0.0522$ |
| DADT (Ours) | **0.0105** $\pm 0.0003$ | **0.0180** $\pm 0.0004$ | **0.0368** $\pm 0.0051$ | **0.0230** $\pm 0.0004$ | **0.0094** $\pm 0.0005$ |

DT, demonstrating the effectiveness of the proposed prediction loss. Moreover, we also compare DT and DADT with Transformer-BC, which does not condition on return-to-go and is trained in a manner of behavior cloning (BC). We find that both DT and DADT significantly outperform Transformer-BC. These results exhibit that return-to-go conditioned training is more capable of processing offline dataset, which contains suboptimal behaviors, more effectively than BC in dynamics generalization.

### 5.4 Jointly optimizing state and action prediction boosts dynamics generalization

To further investigate whether including the proposed auxiliary state prediction loss is helpful for dynamics generalization, we compare our method and alternative training scheme, which also learns state prediction but in a different manner, with respect to next state prediction error in unseen dynamics. In this context, we design DT+, a variant of DT; it is trained using only action prediction loss (as the original DT does), and then the additional linear layer for state prediction is stacked and trained on top of the (frozen) trained transformer. Namely, a comparison between our DADT and DT+ could support how jointly optimizing state prediction loss can help transformer's capabilities of dynamics generalization. As shown in Table 3, DADT exhibits far lower next state prediction error across the environments even if DT+ uses the same amount of data and same objective function. This result further supports that learning next state prediction in joint with action conditioned on desired return is effective in dynamics generalization.

### 5.5 How is DADT compute-efficient?

Figure 3 shows the training throughput of DADT and baselines on Ant environment. The throughput is measured as the number of updated gradient steps per second on a single GeForce RTX 2080 Ti GPU and an Intel Xeon CPU E5-2630 @ 2.80GHz. We find that DADT requires about 6x less time than other baselines. This is because FOCAL has quadratic time complexity in distance metric learning, and MerPO trains separated networks for dynamics model and policy in order, while DADT jointly trains both return-conditioned policy and dynamics information in linear time complexity with single transformer.

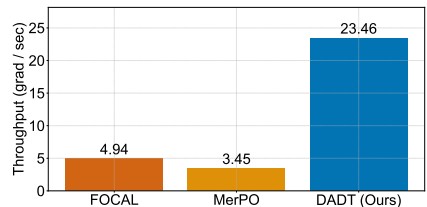

Figure 3: Throughput of DADT and baselines on Ant environment.

### 5.6 Importance of transformer architecture in next state prediction

To further investigate the source of capabilities for dynamics generalization of DADT, we compare next state prediction error of DADT and alternative architectures. For the comparison, we consider three types of architectures: Multi-Layer Perceptron (MLP), LSTM (Hochreiter & Schmidhuber, 1997), and Context-aware Dynamics Model (CaDM) (Lee et al., 2020), which extracts context from history of state-action pairs to predict next state. We remark that MLP only consumes the current state without considering past transitions, while LSTM and CaDM watch a fixed number of past transitions along with recurrent model and ensemble of MLPs, respectively.

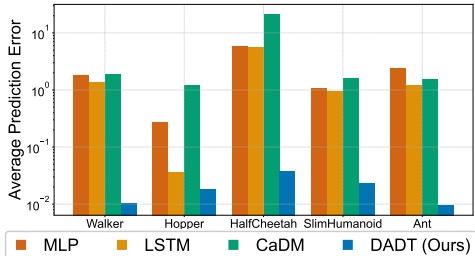

Figure 4: Average next state prediction error on unseen dynamics across 5 runs of DADT and baseline architectures. Y-axis is represented on the log-scale.

As shown in Figure 2, we observe that DADT shows a far low prediction error overall. This result demonstrates that the generalization ability of DADT comes from the high state prediction ability of transformer architecture.

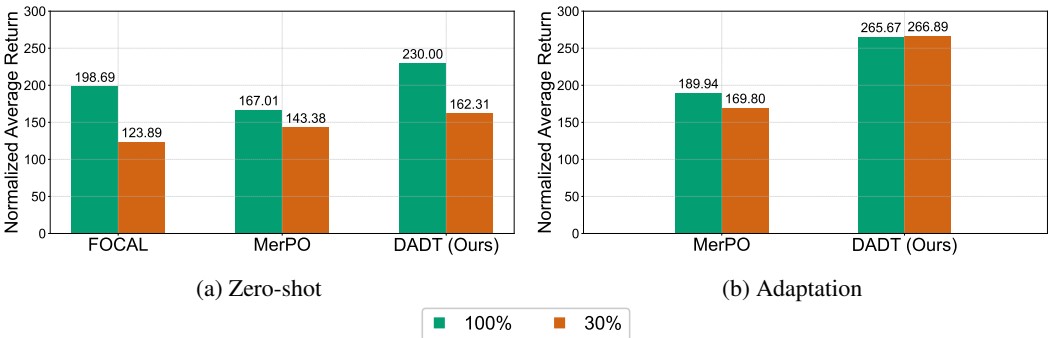

(a) Zero-shot                    (b) Adaptation

■ 100%   ■ 30%

Figure 5: Normalized average return on test dynamics of baseline models and DADT trained with different subset of data. We sort trajectories contained in offline dataset with increasing order of episode returns, and X% denotes how much we use the dataset starting from the trajectory of the lowest return. Namely, in the setting of 30%, models are much likely to be trained using sub-optimal offline dataset in training than the setting of 100%.

### 5.7   How does DADT perform in low data-quality regime compared to baselines?

To investigate how DADT performs in a low data-quality regime, we compare DADT and baselines with a subset of offline dataset , which is consisted of suboptimal trajectories. Specifically, we build a new offline dataset by selecting trajectories of which the average return is in the bottom 30% of the original offline dataset; note that the dataset for adaptation is not re-built. As shown in Figure 5, DADT shows the best performance both in zero-shot and adaptation in the low data-quality regime.

### 5.8   How does balancing coefficient affect performance?

Finally, we investigate how the the balancing coefficient $\lambda$, which determines how state prediction loss occupies total loss, affects the performance of DADT. Figure 4 shows the overall performance in unseen dynamics by aggregating the average return across all environments with different $\lambda$. One can observe that the difference in performance by $\lambda$ is not significant. This result indicates that our performance of DADT is robust to the choice of hyperparameter $\lambda$.

Table 4: Aggregated evaluation result of DADT with varying balancing coefficient $\lambda$.

| $\lambda$ | Average return |
|------|-----------------|
| 0.01 | 45.56 $\pm 19.58$ |
| 0.05 | **46.17** $\pm 20.74$ |
| 0.1  | 45.97 $\pm 20.93$ |
| 0.5  | 45.62 $\pm 19.83$ |
| 1.0  | 44.10 $\pm 19.55$ |

## 6   Conclusion

In this work, we proposed Dynamics-Augmented Decision Transformer (DADT), a simple yet effective transformer-based model for offline dynamics generalization with auxiliary state prediction loss. We have shown that our model outperforms state-of-the-art algorithms in various control tasks. Remarkably, DADT without additional adaptation surpasses fine-tuned baselines in unseen dynamics. We believe that DADT will guide new interesting directions in offline dynamics generalization.

**Limitation.** While our work shows impressive results in offline dynamics generalization with various continuous control tasks, we have considered the limited context of varying dynamics (i.e., mass, friction, and inertia). However, in the real world, agents often encounter a more diverse and difficult situation of dynamics generalization rather than varying dynamics coefficient, i.e., navigating various shapes of mazes. Therefore, it would be an interesting future direction to extend our work into a more complex setting with more practical experimental setups like autonomous driving (i.e., CARLA (Dosovitskiy et al., 2017)). Extending our work into setups where reward function also changes along with varying dynamics would be an interesting future step.

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

# A   Environment details

We use 5 simulated robotic control environments from OpenAI Gym (Brockman et al., 2016). We largely follow the implementation details of the Gym [1] for resetting agent state and shaping reward function of Walker, Hopper, HalfCheetah, and Ant. For SlimHumanoid, we change the environment code of Humanoid in Gym, following the description in (Wang et al., 2019).

**Walker.** Walker is a two-dimensional two-legged figure with 7 rigid links, including a torso and 2 legs. The goal is to move forward as fast as possible while maintaining the standing height and consuming as small control input as possible.

- Observation. Observation is a 17-dimensional vector that includes 1) root joint's position, 2) angular position and velocities.

- Action. $a \in [-1.0, 1.0]^6$ represents torques applied at six joints connecting the six body parts.

- Reward. $r_t = \dot{x}_{torso,t} - 0.001 \|a_t\|^2$ where $\dot{x}_{torso,t}$ represents forward velocity of the torso. We also add an alive bonus of 1 to the agents at every time-step.

**Hopper.** Hopper is a two-dimensional one-legged figure consisting of 4 rigid links, including a torso, thigh, leg, and foot. The goal is to move forward as fast as possible while maintaining the standing height and consuming as small control input as possible.

- Observation. Observation is a 11-dimensional vector that includes the angular position and velocity of all the joints, except for the $x$ position of the root joint.

- Action. $a \in [-1.0, 1.0]^3$ represents torques applied at three joints connecting the four body parts.

- Reward. $r_t = \dot{x}_{torso,t} - 0.001 \|a_t\|^2$, where $\dot{x}_{torso,t}$ represents forward velocity of the torso. We also add an alive bonus of 1 to the agents at every time-step.

**HalfCheetah.** HalfCheetah is a two-dimensional two-legged figure. It consists of 7 rigid links (1 for torso, 3 for forlimb,, 3 for hindlimb). It is connected by 6 joints, to which an actuator is attached per each joint. The goal is to move forward as quickly as possible while minimizing the cost for controlling agent as small as possible.

- Observation. Observation is given by a 17-dimensional vector that includes 1) root joint's position (except for the x-coordinate) and velocity, and 2) center of mass of the torso.

- Action. $a \in [-1.0, 1.0]^6$ represents torques applied at six joints.

- Reward. $r_t = \dot{x}_{torso,t} - 0.1 \|a_t\|^2$, where $\dot{x}_{torso,t}$ represents forward velocity of the torso.

**SlimHumanoid.** Humanoid is a 3D robot for simulating a human. It has a pair of legs and arms unlike other environments, which is consisted of 13 rigid links, 17 actuators. The goal is to move forward as quickly as possible while minimizing the cost for controlling agent as small as possible. SlimHumanoid Wang et al. (2019) is a variant of Humanoid with limited observation and ease of use. Specifically, there are some changes in observation details (excluding the center of mass-based quantities, external force, and actuator force) and reward function (no penalizing on external force).

- Observation. Observation is a 45-dimensional vector that includes angular position and velocities.

- Action. $a \in [-0.4, 0.4]^{17}$ represents torques applied at seventeen joints.

- Reward. $5/4 \times \dot{x}_t - 0.1 \|a_t\|_2^2 + 5 \times bool(1.0 <= z_t <= 2.0)$, where $\dot{x}_{torso,t}$ represents forward velocity of the torso, and $z_t$ is the height of the torso. We also add an alive bonus of 5 to the agents at every time-step.

---

[1] https://github.com/openai/gym

**Ant.** Ant is a 3D ant-like robot consisting of a torso with 4 legs, each of which is connected with 2 joints. There are 8 actuators consisting of 13 rigid links. The goal is to move forward as fast as possible while maintaining the standing height and consuming as small control input as possible.

- Observation. Observation is a 27-dimensional vector that includes angular position and velociyu of all 8 joints, except for the $x$ and $y$ positions of the root joints.

- Action. $a \in [-1.0, 1.0]^8$ represents torques applied at eight joints connecting the two links of each leg and torso.

- Reward. $r_t = \dot{x}_{torso,t} - 0.5\|a_t\|_2^2 - 0.5 \times 0.001\times + 1.0$, where $\dot{x}_{torso,t}$ represents forward velocity of the torso. We also add an alive bonus of 1 to the agents at every time-step.

# B  Offline dataset details

**Dataset Collection.** For each environment, we collect 20 train dynamics and 5 test dynamics by changing the parameter of the agent, specifically body mass, body inertia, damping, and friction. Parameters of each dynamics are randomly chosen by multiplying coefficient (1) $1.5^r, r \sim \mathtt{Uni}(-3, 3)$ for mass, inertia, friction and (2) $1.3^r, r \sim \mathtt{Uni}(-3, 3)$ for damping with initial values.

We train policies using SAC Haarnoja et al. (2018) for every single dynamics. We skipped the first 50K steps for warm-up. For collecting datasets with diverse qualities, we collect 50 random rollouts from trained policies every 50K timesteps. The total dataset size of each environment is from 1.3GB (Hopper) to 1.7GB (Ant).

You can download the dataset we used for experiments in the link below[2].

**Normalization.** Return statistics of the collected offline trajectories can be found in Table 5. We used the minimum and maximum value among test dynamics of each environment for normalizing the average return in evaluation.

Table 5: The return statistics of offline trajectories for test dynamics. We mark the minimum/maximum scores of each environment used for normalization to be bold. We used scores marked to be bold for normalizing average return.

| | Walker | | | | |
| --- | --- | --- | --- | --- | --- |
| | Dynamics 20 | Dynamics 21 | Dynamics 22 | Dynamics 23 | Dynamics 24 |
| min | -15.56 | 123.40 | **-17.73** | -13.32 | 52.18 |
| max | 645.11 | **769.63** | 674.81 | 725.99 | 645.11 |
| mean | 519.47 | 548.16 | 496.15 | 514.39 | 519.47 |
| | Hopper | | | | |
| | Dynamics 20 | Dynamics 21 | Dynamics 22 | Dynamics 23 | Dynamics 24 |
| min | 129.77 | 10.72 | **8.66** | 12.70 | 15.53 |
| max | 678.26 | **754.62** | 710.44 | 668.36 | 682.38 |
| mean | 537.53 | 605.03 | 555.52 | 542.72 | 615.87 |
| | HalfCheetah | | | | |
| | Dynamics 20 | Dynamics 21 | Dynamics 22 | Dynamics 23 | Dynamics 24 |
| min | 149.69 | 69.57 | 168.52 | **31.33** | 71.92 |
| max | 1681.41 | 2032.87 | 1484.98 | 1419.93 | **2248.80** |
| mean | 1222.03 | 1412.65 | 1281.07 | 1003.75 | 1686.36 |
| | SlimHumanoid | | | | |
| | Dynamics 20 | Dynamics 21 | Dynamics 22 | Dynamics 23 | Dynamics 24 |
| min | 148.34 | 147.26 | 128.95 | 106.63 | **102.19** |
| max | 975.08 | 1061.81 | 1107.90 | **1261.80** | 1217.33 |
| mean | 444.97 | 461.60 | 495.55 | 512.27 | 537.16 |
| | Ant | | | | |
| | Dynamics 20 | Dynamics 21 | Dynamics 22 | Dynamics 23 | Dynamics 24 |
| min | -61.30 | -29.38 | -27.31 | -60.95 | **-68.82** |
| max | 786.27 | 910.21 | 968.01 | 744.02 | **1354.02** |
| mean | 348.11 | 494.00 | 431.37 | 255.81 | 858.21 |

---

[2]`https://drive.google.com/file/d/1vWoceRc8eFzCNdTUHe44w2Cgreeru9NV/view?usp=sharing`

# C  Implementation details

In this section, we will explain the implementation details and resources used for training and evaluating DADT. Specifically, we train DADT for 1M steps using AdamW optimizer (Loshchilov & Hutter, 2017) with a learning rate of $3 \times 10^{-5}$ containing linear warmup steps of 10K, weight decay of $10^{-4}$, gradient clip of 1.0, and batch size of 64. For fine-tuning DADT on unseen dynamics, we train DADT with a learning rate of $10^{-5}$, weight decay of $10^{-2}$, and batch size of 32. We include the code of DADT in the supplementary material.

**Evaluation Protocol** To facilitate comparison across different environments, we normalize returns for each dynamics to the range between 0 to 100, by computing `Normalized_return` $= 100 \times \frac{\texttt{return} - \texttt{random\_return}}{\texttt{expert\_return} - \texttt{random\_return}}$ following setups in Chen et al. (2021) and Fu et al. (2020). For `random_return` and `expert_return`, we use minimum/maximum return of collected trajectories, respectively, which are collected periodically while training a soft actor-critic (Haarnoja et al., 2018) for each test dynamics. Finally, we average out 5 (runs) $\times 5$ (test dynamics) $\times 10$ (rollouts) number of returns to measure performance. For DADT and baselines, we measure the test performance periodically during training and report the best return.

**Hyperparamters.** Hyperparameters for DADT are shown in Table 6. We choose target return-to-go, referring to expert performance for each environment, which is set to a value equivalent to or slightly above expert performance. We heuristically find that training (1) larger DADT and (2) DADT with more gradient steps boost the model's performance but use the current version for faster training and evaluation.

Table 6: Hyperparameters of DADT.

| Hyperparameter | Value |
|---|---|
| Number of layers | 3 |
| Number of attention heads | 1 |
| Embedding dimension | 128 |
| Batch size | 64 |
| Context length $K$ | 20 |
| Return-to-go conditioning | 3000 for HalfCheetah |
| | 1000 for Walker, Hopper, SlimHumanoid, Ant |
| Dropout | 0.1 |
| Learning rate | $3 \times 10^{-5}$ |
| State prediction weight $\lambda$ | 0.05 |
| Optimizer | AdamW (Loshchilov & Hutter, 2017) |
| Optimizer Momentum | $\beta_1 = 0.9, \beta_2 = 0.99$ |
| Weight decay | $10^{-4}$ |
| Warm-up steps | 10K |

**Resources.** For training and evaluating our model, we use a single NVIDIA GeForce RTX 2080 Ti GPU and 40 CPU cores (Intel(R) Xeon(R) CPU E5-2630 @ 2.80GHz), taking at most 8 hours for training and less than 10 seconds for rollout 10 trajectories per evaluation. Compared to the original DT (Chen et al., 2021), DADT shows no significant increase in training time.

# D  Comparison with Trajectory Transformer

While our next state prediction loss has a connection with that of Trajectory Transformer (Janner et al., 2021) in that both of them predict next state given past transitions, we remark that DADT does not use discretization, which might be a negative factor in dynamics generalization. Specifically, approximating a set of values from different dynamics using the exact discrete quantities could hurt capturing subtle changes in distinguishing the difference between dynamics.

As table 7 demonstrates, our DADT consistently outperforms TT. We think that the underperforming of TT is caused by (1) reduced visible context due to per-dimension processing and (2) information loss from discretization.

Table 7: Normalized average return on test dynamics with episode length 200 across 5 runs. We mark the scores within one standard deviation from the highest average score to be bold.

| Environment | Evaluation | FOCAL (Li et al., 2020) | MerPO (Lin et al., 2022) | TT (Janner et al., 2021) | DADT (Ours) |
|---|---|---|---|---|---|
| Walker | Zero-shot | $52.44 \pm 17.31$ | $40.67 \pm 6.66$ | $45.76 \pm 12.08$ | $\mathbf{61.12} \pm 4.99$ |
| | Adaptation | - | $48.11 \pm 6.95$ | $32.74 \pm 3.3$ | $\mathbf{64.65} \pm 4.42$ |
| Hopper | Zero-shot | $41.56 \pm 12.44$ | $53.21 \pm 1.89$ | $55.22 \pm 6.42$ | $\mathbf{74.69} \pm 4.22$ |
| | Adaptation | - | $57.38 \pm 6.24$ | $15.26 \pm 6.26$ | $\mathbf{83.11} \pm 3.28$ |
| HalfCheetah | Zero-shot | $10.60 \pm 4.10$ | $8.06 \pm 1.09$ | $9.88 \pm 1.80$ | $\mathbf{16.29} \pm 1.95$ |
| | Adaptation | - | $14.36 \pm 3.27$ | - | $\mathbf{23.48} \pm 3.39$ |
| SlimHumanoid | Zero-shot | $20.45 \pm 1.60$ | $33.73 \pm 1.41$ | $20.12 \pm 1.07$ | $\mathbf{42.03} \pm 1.68$ |
| | Adaptation | - | $32.41 \pm 1.52$ | - | $\mathbf{39.58} \pm 1.79$ |
| Ant | Zero-shot | $18.73 \pm 1.28$ | $31.34 \pm 0.80$ | $14.79 \pm 1.85$ | $\mathbf{35.87} \pm 1.62$ |
| | Adaptation | - | $37.08 \pm 0.79$ | - | $\mathbf{54.85} \pm 1.81$ |

