# OpenReview forum: "Dynamics-Augmented Decision Transformer for Offline Dynamics Generalization"
_NeurIPS.cc/2022/Workshop/Offline_RL — Offline RL Workshop NeurIPS 2022_

### Official Review · Reviewer_S3z2 · 2022-10-18
**An interesting paper, but without detailed explanation on the model**

**Rating:** 5
**Confidence:** 4

**Review:**

In this paper, the authors propose a transformer-based model, aka DADT, to tackle the problem of offline-dynamics generalization. My whole feeling is that this paper is kind of an experimental report only. The main concern I have is that the authors seemingly do not put much effort into explaining why the proposed transformer-based model works for offline dynamics generalization. Because from the description of the DADT architecture in Section 2, it seems hard to get any clue about the reasons. Without such insights, the proposed approach would be of limited use to the community. I suggest the authors devote more time in the updated version to exploring why and how the DADT architecture works in detail.